



# Basal traction mainly dictated by hard-bed physics over grounded regions of Greenland

Nathan Maier[1], Florent Gimbert[1], Fabien Gillet-Chaulet[1], and Adrien Gilbert[1]

[1]University of Grenoble Alpes, CNRS, IGE, Grenoble, France

*Correspondence to*: Nathan Maier (ntmaier@gmail.com)

**Abstract.** On glaciers and ice sheets, identifying the relationship between velocity and traction is critical to constrain the bed physics that control ice flow. Yet in Greenland, these relationships remain unquantified. We determine the spatial relationship between velocity and traction in all eight drainage catchments of Greenland. The basal traction is

estimated using three different methods over large grid cells to minimize interpretation biases associated with unconstrained rheologic parameters used in numerical inversions. We find the relationships are consistent with our current understanding of basal physics in each catchment. We identify catchments that predominantly show Mohr-Coulomb-like behavior typical of deforming beds or significant cavitation, as well as catchments that predominantly show rate-strengthening behavior typical of Weertman-type hard-bed physics. Overall, the traction relationships

suggest that the flow field and surface geometry over the grounded regions of the Greenland ice sheet is mainly dictated by Weertman-type hard-bed physics. Given the complex basal boundary across Greenland, the relationships are captured surprisingly well by simple traction laws over the entire velocity range, including regions with velocities over 1000 m/yr, which provide a parameterization that can be used to model ice dynamics at large scales. The results and analysis serve as a fundamental constraint on the physics of basal motion in Greenland and provide unique insight

into future dynamics and vulnerabilities in a warming climate.

## 1 Introduction

For glaciers and ice sheets the relationship between basal motion, basal traction, and the response to external forcing are fundamental to realistic ice flow modeling. The physics that control the interdependencies between these characteristics is dictated by the properties of the bed. For hard beds basal traction is related to the viscous drag

generated as ice slides around bed roughness (Weertman, 1964). Meltwater modulates friction and sliding over hard beds by occupying cavities on the lee-side of bedrock bumps (Gagliardini et al., 2007; Lliboutry, 1968; Schoof, 2005), increasing sliding by reducing the apparent roughness of the bedrock. For deformable till beds the traction and occurrence of basal motion primarily depends on the failure strength of the till which acts as a Mohr-Coulomb material (Iverson et al., 1998; Kamb, 1991; Tulaczyk, 1999; Zoet and Iverson, 2020). Meltwater can weaken deformable beds

by increasing pore pressures within the till (Iverson et al., 1998, 2003) and induce bed deformation where there was previously none.



Constraining the type of bed is critical to understanding the dynamic response to future forcing given the difference in the physical processes that control ice motion (Bougamont et al., 2014). In Greenland, the composition of the basal substrate is known to vary significantly. There are both hard and till beds (Booth et al., 2012; Cooper et al., 2019;

Dow et al., 2013; Doyle et al., 2018; Harper et al., 2017; Kulessa et al., 2017; Lindbäck and Pettersson, 2015; Lüthi et al., 2002). The exact distribution of each is unknown, but direct observations have shown that they coexist at the regional scale (Booth et al., 2012; Dow et al., 2013; Harper et al., 2017). Hydrologic regimes also vary significantly. In low-elevation regions meltwater is delivered to the bed via supraglacial routing of water into moulins (Smith et al., 2015), while at higher elevations water stored in supraglacial lakes can be suddenly injected into the basal interface

during catastrophic lake drainages (Das et al., 2008; Stevens et al., 2015). In other regions, basal melting from high geothermal heat fluxes provides an in situ source of basal water independent of surface melt (Fahnestock et al., 2001).

Given complexity of the basal boundary in Greenland, the roles of different physical processes operating at the base in controlling sliding speeds and surface geometry of the ice sheet at large scales have historically been difficult to quantify. Improvements over the last decade in satellite observations of surface velocity and geometry have made it

possible to infer bed properties based on ice dynamics (Gillet-Chaulet et al., 2016; Habermann et al., 2013; Minchew et al., 2016), bypassing the need for localized observations at the basal interface to characterize the bed. The main approach is to use time variations in the ice flow and surface geometry to constrain characteristic properties of different bed types (Gillet-Chaulet et al., 2016; Habermann et al., 2013). This approach has been used to infer the apparent sliding exponent at Pine Island Glacier and Antarctica (Gillet-Chaulet et al., 2016), and changes in bed strength

underneath Jakobshavn (Habermann et al., 2013). However, using time variations to constrain bed properties is limited to regions where large changes in the surface speeds and ice geometry coincide and cannot be applied to much of the ice sheet where dynamic changes are small.

An alternative way to infer bed properties is to use the spatial distribution of velocity and traction to determine the physical relationship between the variables. A study on Hofsjökul, a small isothermal ice cap located in Iceland, used

inverted tractions and sliding velocities to show that the relationship between velocity and traction is consistent with deforming bed physics (Minchew et al., 2016). Yet explicit inverting for basal traction using a shallow approximation or full stokes representation of the momentum balance yields a traction solution that is non-unique and is dependent on the a priori choice of model parameters including both physical parameters related to ice rheology and parameters within regularization schemes necessary to stabilize the numerical solvers and avoid data overfit (Habermann et al.,

2013; Shapero et al., 2016). The influence of the parameters on traction solution are difficult to evaluate, but are expected to be particularly important where rheologic parameters are poorly constrained which can lead to biases in the interpretations of the bed conditions (Habermann et al., 2013; Joughin et al., 2012).

The non-linear ice rheology dictated by "Glen's Law" is only known to an approximate degree, but is known to have strong dependencies on ice temperature (Cuffey and Paterson, 2010). The Greenland Ice Sheet has a complex

polythermal structure especially near the margin where cryohydrologic warming is pervasive (Harrington et al., 2015; Lüthi et al., 2015). Further, in Greenland the uncertainty of the deformation exponent, $n$, is large. Direct estimates of



$n$ range considerably from 2.5 to 4.5 (Bons et al., 2018; Dahl-Jensen and Gundestrup, 1987; Gillet-Chaulet et al., 2011; Lüthi et al., 2002; Ryser et al., 2014). Other aspects of ice flow such as crevasses that are common in regions of concentrated fast flow (Cavanagh et al., 2017; Lampkin et al., 2013) and in extensional regimes (Poinar et al.,
2015), are not accounted for in high-order inversions but could significantly impact the relationship between the velocity field and higher-order stresses (stress gradients). Together, these aspects make it difficult to confidently estimate the rheology to invert for tractions across the Greenland Ice Sheet and use the spatial relationship between basal traction and velocity to infer bed properties. An initial study found no relationship between velocities and tractions for the fast-flowing outlet glaciers located on the periphery of Greenland (Stearns and van der Veen, 2018).
While this result is still under debate (Minchew et al., 2019; Stearns and van der Veen, 2019), a poorly constrained rheology may have had an impact on their findings.

In this study, we compare multi-year averaged surface velocities to the basal traction estimated using three different methods over large 6 km x 6 km grid cells (*Figure 1*) to determine the form of the velocity – traction relationship within all eight drainage catchments in Greenland (Zwally et al., 2012). Our approach to infer the basal traction is
designed to minimize interpretation bias associated with parameter choice in numerical inversions. Averaging the traction across large length scales diminishes the role of higher-order stresses and thus the need to correctly prescribe the ice rheology to invert for basal traction. Using the three inversions with different complexity (Shallow Ice Approximation, Shallow Stream Approximation, and Full Stokes), we can determine the length scale at which the influence of higher order stresses is minimized. The method also separates topographic-scale form drag, which arises
from ice flowing around large scale topography (Kyrke-Smith et al., 2018), from the basal traction due to skin drag, which is the component of the basal resistance dictated by bed properties that are of interest for traction laws.  Where the three inversions converge, we can confidently infer the form of the velocity and traction relationship, and where they diverge, we can constrain the range of basal physics possible given the different assumptions within each inversion method. We compare the form of the velocity-traction relationships to expectations given different bed
physics to interpret the dominant basal processes operating at the catchment scale. Finally, we discuss the implications of the results for current and future ice dynamics in Greenland.

## 2 Methods

### 2.1 Data

All three inversions use identical surface elevation, bed elevation, and surface velocity data sets (*Figure 1*). We use
the GIMP digital elevation model of the ice sheet surface (30 m resolution) (Howat et al., 2014, 2017) and ice thicknesses and bed topography from BedMachine v3 (150 m resolution) (Morlighem, 2018; Morlighem et al., 2017). We use surface velocities from a multi-year velocity mosaic (250 m resolution) with complete coverage of Greenland (Joughin et al., 2016, 2018) and low absolute errors (generally < 3 m/yr). The velocities represent decadal-scale averages generated from data collected from 1995 – 2015 but are mostly derived from data from 2005 – 2015. The
time averaging of velocities incorporates both summer and winter flow, and thus they are not expected to reflect a strong seasonality (Joughin et al., 2018). To ensure high data fidelity we exclude regions where the error and



uncertainty in either the velocity or driving stress exceeds 50% of its magnitude (supporting *Figure S5*). We restrict our analysis to thawed regions of each catchment where basal motion is expected to occur. This is delineated using a map of the basal thermal state (*Figure 1*) (MacGregor et al., 2016, 2017).

**2.2 Basal Traction Inversions**

**2.2.1 Shallow Ice Approximation**

The Shallow Ice Approximation (SIA) neglects higher-order stresses and thus assumes that the basal traction as equivalent to the gravitational driving stress (Morland and Johnson, 1980). The SIA is valid when the influence of stress-gradient coupling, which arises from spatially complex flow, is negligible, i.e. typically across length scales of

roughly 10 ice thicknesses (Cuffey & Paterson, 2010; Kamb & Echelmeyer, 1986), and also where the effects of topographic scale form drag are small, i.e. where the bed is mostly flat. While simple, the SIA does not require the use of unmeasured parameters for either the ice rheology or regularization schemes.

We calculate the driving stress across 6 km length scales where the effect of higher-order stresses is minimized using:

$$\tau_d = \rho_i gH tan\alpha = \tau_{b,SIA}, \tag{1}$$

where $\rho_i$ is the density of ice (900 kg/m$^3$), $H$ is the ice thickness, $g$ is the gravitation constant (9.81 m/s$^2$), and $\alpha$ is the surface slope, which is calculated by taking the gradient of the surface elevations. Prior to the calculation, data sets are filtered with a 6 km x 6 km Gaussian kernel with a sigma optimized to minimize the difference between the SIA and SSA (Shallow Stream Approximation) estimates of traction. The choice of length scale is discussed further in *2.3*. We decimate the driving stress and velocity data sets to 6 km grid spacing to generate independent cells. We scale the

estimates of driving stress with the surface area of the bed in each grid cell to account for topographic variations which are needed to estimate skin drag (i.e. tangential tractions). This is equivalent to a shape factor commonly used for alpine glaciers (Cuffey and Paterson, 2010).

**2.2.2 Shallow Stream Approximation**

The Shallow Stream Approximation (SSA) solves the vertically integrated form of the momentum balance in two

dimensions which accounts for higher-order stresses that arise from gradients in the velocity field assuming shearing of the ice column is negligible (MacAyeal, 1989). This approximation is most valid for fast ice flow where shearing is limited or concentrated near the base. The method also implicitly assumes that interactions with the bed besides tangential tractions are negligible and does not explicitly account for topographic-scale form drag. The SSA is therefore most appropriate in regions where the bed topography is relatively flat.

The SSA inversion was implemented in Elmer/Ice, a three-dimensional finite-element ice modeling software (Gagliardini et al., 2013; Goelzer et al., 2017). The inversion domain consists of a triangular element mesh with resolution ranging from 500 m to 5 km across the ice sheet domain (*Figure 1*). The model uses an ice rheology with





a vertically averaged viscosity generated from ice temperatures from a paleo-spin-up of the SICOPOLIS model and $n$
= 3 (Goelzer et al., 2017; Seddik et al., 2012). The basal traction was inverted using the control-inverse method with

a linear sliding law to find the "effective" friction coefficient which optimized to minimize the misfit between the
observed and modeled velocity (Gagliardini et al., 2013; Morlighem et al., 2010). This friction coefficient includes
all the physical dependencies between sliding and tractions. While a linear sliding law is used for the inversion, the
traction field is shown to have little sensitivity to the choice of sliding law (Joughin et al., 2004).

### 2.2.3 Full Stokes Inversion

The Full Stokes (FS) continuum equations represent the full momentum balance without simplifying assumptions.
This means the traction inversions are made in full consideration of vertical shearing of the ice column, higher-order
stresses, and topographic form drag as the ice flows around landscape-scale roughness.

The FS inversion is also implemented in Elmer/Ice on a variable-resolution triangular element mesh with fine
resolution (~150 m) near the margin and coarse resolution within the interior (~ 60 km) (Gagliardini et al., 2013;

Goelzer et al., 2017).The large grid dimensions in the interior reduce the high computation costs of the FS, which
allocates more resources near the margin where ice flow is more complex (*Figure 1*). The ice rheology is calculated
using ice temperatures from the SICOPOLIS paleo-spin-updated for use in ISMIP6 and $n$ = 3 and the basal traction
was inverted using the same methods as the SSA. For consistency with the other inversions, we compare the FS
traction field against the surface velocity in our analysis and instead of the inverted sliding velocities. The results are

very similar in both cases and do not change the interpretations presented in the manuscript (see *supporting
information*).

### 2.3 Length scale for grid cells

We compare the traction estimates for grid cells with different dimensions to determine at what length scale the effects
of higher-order stresses are small, and correspondingly the biases associated with rheology choice in the numeric

inversions are also small. To do so we compare SIA traction computed across grid cells with dimensions of 1.5 – 12
km to the traction from the SSA averaged across the same grid cells (*Figure 2*). We use SSA instead of the FS inversion
as it does not explicitly include bed interactions beyond traction, thus differences in the comparison to the SIA traction
estimates can be attributed solely to higher-order stresses. We find that averaged over length scales greater than 6 km,
the mean difference between the two traction estimates on average is less than 0.015 MPa (Figure 2) which on average

is ~17% of the traction. Increasing the length scale beyond 6 km does further decrease the difference between the data
sets, however, does not change the data interpretations presented in the manuscript, which we note are also similar if
a 3 km length scale is used (*supporting Figure S1, S3*). We choose to use 6 km grid cells in the analysis in order to
maximize the resolution near the margins, lengthen the span of velocity and traction range used to determine the
relationship between the variables, and provide more interpretable relationships in catchments 3 and 5 which have

fewer grid cells. Like the SSA, the FS tractions are averaged across the 6 km grid cells for this analysis. Due to the



coarse nature of the interior grid cells for the FS simulation, we only use grid cells where velocities are greater than 100 m/yr which conservatively delineates where the mesh resolution is less than the 6 km limit (*Figure 1*).

### 2.4 Interpreting the Velocity-Traction Relationships

### 2.4.1 Physical Traction Relationships

The theoretical relationship between velocity and traction for idealized beds have characteristic features that can be used to distinguish between different types of basal physics. Below we outline three types of beds and the diagnostic features that we will use to aid our characterization of the basal physics in each catchment.

*Hard bed (Weertman 1964)*: Weertman sliding describes the velocity-traction relationship for ice sliding around bed roughness without cavitation:

$$\tau_b = \left(\frac{u_b}{A_s}\right)^{1/m},$$   (2)

where $A_s$ is a friction factor that incorporates the basal roughness characteristics as well as the near basal ice rheology, $u_b$ is the sliding velocity, and $m$ is the sliding exponent. This is a rate-strengthening relationship where increasing sliding result in increasing stress (*Figure 3*).

*Hard bed with cavitation (Gagliardini 2007)*: A hard bed relationship that includes cavitation combines aspects of
both rate-strengthening and Mohr-Coulomb-like behavior given by:

$$\tau_b = CN\left(\frac{X}{1+\alpha X^q}\right)^{1/n},$$   (3)

where $\alpha$ is a function of $q$ and $X = \frac{u_b}{C^n N^n A_s}$. Here $q$ is a bed shape parameter, $C$ a parameter related to the steepness of the bed roughness, $N$ is the effective pressure at the bed, $A_s$ is friction factor comparable to that in the Weertman relation, and $u_b$ is the sliding velocity. Here, rate strengthening occurs at high effective pressures producing a
relationship identical to Weertman sliding (*Eq. 2, Figure 3*). When $CN$ is approached the curvature of the relationship begins to increase, deviating from Weertman behavior. This continues until $CN$ is reached which is commonly known as Iken's bound (Iken, 1981).  Further increases beyond Iken's bound would cause the basal drag to decrease (rate-weakening) causing the ice flow to unstable in the absence of higher-order stresses. However, outside of rare instances leading to glacier collapse (Kääb et al., 2018), glaciers remain stable because sliding and traction in this range is
controlled non-locally by higher-order stresses.

*Deformable Bed (Zoet 2020)*: Traction for deformable beds combines rate strengthening and Mohr-Coulomb behavior. Below the strength of the till basal motion occurs as over a hard bed (same as described in *Eq. 2*), occurring at the interface between the ice and the till as the ice deforms around bed roughness. Once the basal traction reaches the





yield stress ($\tau_*$) till deformation begins. Further increases in stress are limited and the velocity becomes independent of the stress, which like with cavitation, is dictated non-locally by higher-order stresses. This produces a characteristic sharp kink in the velocity traction relationship at the yield stress of the till (*Figure 3*). Over the large grid cells used in this analysis the gravitational stress is mainly balanced by the basal traction. For grid cells with mostly deforming beds this means higher-order stresses will only play a minor role in the stress equilibrium but ice flow in the surrounding regions will still control the velocity field. This same logic applies for the rate-weakening range for hard beds with significant cavitation.

### 2.4.2 Deformation Motion

The surface velocity reflects the combined motion due to basal motion and internal ice deformation and thus interpretation of the derived relationships requires consideration of both components of motion. In Greenland, the rheologic uncertainty and complex thermal structure make it difficult to infer a location specific rheology to confidently calculate deformation across the ice sheet (Maier et al., 2019). However, direct measurements of basal motion collectively show the prevalence of high fractions of basal motion [0.44 – 0.96] across a wide range of glaciological environments in Greenland (Doyle et al., 2018; Lüthi et al., 2002; MacGregor et al., 2016; Maier et al., 2019; Ryser et al., 2014).

Nevertheless, we demonstrate the relationship between surface velocity and traction remains interpretable in terms of bed properties even if deformation motion contributes significantly to the surface velocity and/or the parameters that controls the rate of deformation change significantly towards the margin. In *Figure 3* we add deformation motion to the traction relationships presented in *2.4.1* for a temperate ice column with a thickness of 2500 m using:

$$u_d = \frac{2A(T)}{n+1}\tau_b^n H, \tag{4}$$

where $A(T)$ is a temperature dependent rate factor (calculated following Cuffey and Paterson 2010), $n$ is the flow law exponent equal to 3, and $\tau_b$ is the basal traction as dictated by the relationships in *Figure 3*. This corresponds to a deformation fraction of 0.33 for the rate-strengthening range of the traction relationships, which is similar to that measured along the margins (Doyle et al., 2018; Lüthi et al., 2002; MacGregor et al., 2016; Maier et al., 2019; Ryser et al., 2014), and would approximate deformation where changes in flowline ice thickness are small, i.e. some trough-bound outlet glaciers. We show the characteristic features of the relationship are identical as to the theoretical relationships which relate basal motion to traction. We then test how systematic changes in deformation motion due to marginal thinning or ice warming, which is documented as ice moves from the interior to the margin (Harrington et al., 2015; Lüthi et al., 2015), could influence the traction relationship. We find that even with large marginal ice thickness (3000 – 200 m) and ice temperature changes (-25° – 0°) occurring through the velocity range, the characteristic features of the traction relationships remain unchanged. The persistence of the traction relationships with added deformation motion arises from the strong and non-linear dependence of deformation motion on the basal





traction which is controlled by the properties of the bed. Thus, we can interpret the characteristic features that define each type of bed-physics using surface velocities which remain clearly distinguishable even with deformation motion.

### 2.4.3 Power-law model fitting

We fit a general flow approximation to the observed surface velocity ($u_s$) and traction range in each catchment using:

$$\tau_b = C_p u_s^{\frac{1}{p}},$$    (5)

where $C_p$ is a rate parameter and $p$ is an apparent flow exponent. Since the surface velocity reflects the combined motion due to internal ice deformation and basal motion this equation approximates the combined physics and nonlinearities of each flow mechanism. However, as shown in *Figure 3* the main features of the traction relationships, and therefore the model fit, will be dictated by the basal physics.

We use the magnitude of $p$ as an additional indicator of basal physics and hydrologic influence to interpret catchment dynamics (Joughin et al., 2019). Frozen beds are expected to have $p$ approximately equal to the deformation exponent $n$. The regelation and creep processes (Cuffey and Paterson, 2010; Weertman, 1964) that dictate hard bed sliding suggest $m \approx 2.5 - 3$ for $n = 3$, which is similar to that expected over frozen beds. However, when cavities form on the lee side of bedrock bumps as in *Eq. 3*, the apparent sliding exponent $m$ and thus $p$ have the potential to increase

beyond this range due to decreased ice-bed contact area (Gagliardini et al., 2007; Joughin et al., 2019), which increases curvature in the velocity-traction relationship (Gagliardini et al., 2007; Joughin et al., 2019). For deforming beds $p$ is expected to be large to approximate Mohr-Coulomb behavior (Iverson et al., 1998; Kamb, 1991; Tulaczyk et al., 2001). This is supported by in-situ field (Gillet-Chaulet et al., 2016) and laboratory (Kamb, 1991) experiments which show $m \geq 20$ over deforming beds.

We fit the model to the binned velocity – traction relationship in the thawed and frozen regions of each catchment (number of bins = 20, logarithmically spaced) using a nonlinear least-squares regression in *MATLAB*. Fitting the binned data gives equal weights to fast flowing and slow-flowing regions of each catchment regardless of the amount of grid cells of each. We do not fit models to the FS tractions which we limit to velocities above 100 m/yr and do not reflect the full velocity range. We note that velocity and traction magnitudes are dampened due grid cell averaging,

thus the binned-relationships and model fits will reflect this range rather than the full range possible in each catchment.

### 3. Results

### 3.1 Relationship between velocity and basal traction

The velocity versus traction relationships have a large degree of scatter but show increasing velocities result in increasing traction (*Figure 4*). The binned relationships (magenta dots) are visually well defined in all catchments and

for each estimate of traction and show good agreement with the model fit at both high and low velocities. The high $R^2$



[0.22 – 0.99] indicate that the power-law model generally describes most of the variability of the binned relationship in each catchment (*Figure 4, supporting Table S1*) for the SIA and SSA tractions. With the exception of catchment 2, 3 and 5, the interquartile range (IQR) for each bin occupies a small region of the total data spread indicating the binned relationships represents a large fraction of the data in each catchment.

The relationships for SIA, SSA, and FS inversions show varying degrees of consistency in each catchment (*Figure 5*). In general, the different inversion methods show the same traction variations through the velocity range but show some offset in the basal traction. Bins with offset between the SIA and SSA suggest higher-order stresses may influence tractions, while bins with offset between the SSA and FS suggest topographic scale roughness may support some of gravitation stress through form drag. All catchments show offset between the SIA and SSA at some bins,

however, the offset is small (generally < 0.02 MPa). Similarly, catchments show some differences between SSA and FS. In catchments 6 through 8, we find the relationships increasingly diverge as velocities increase, where peak separation between the SIA, SSA, and FS coincides with the highest velocities.

The binned velocity-traction relationships (*Figure 5*) all have one or more of the characteristic features found in the theoretical traction relationships presented in *Figure 3*. In catchment 1 and 2 there is rate-strengthening until a visible

kink in the traction relationships at ~0.1 MPa and ~0.08 MPa respectively where tractions level off. In catchment 2 this is followed by a large dip in the traction relationship where tractions decrease to a local minimum at ~230 m/yr. In both Catchment 1 and 2, the highest velocities again correspond with high tractions which fall on a similar trend as the binned relationship before the kink. Catchments 3 and 4 show rate-strengthening through the entire velocity range. Catchment 5 shows rate-strengthening until ~ 40 m/yr where tractions abruptly dip until ~60 m/yr. The SIA and FS

then begin to increase again at a similar rate as before the dip but offset towards higher velocities, while the tractions for the SSA level off at ~ 0.095 MPa. Catchment 6 predominantly shows rate strengthening, with a small kink at ~50 m/yr which corresponds with an increase in data scatter and IQR span. Catchment 7 also predominantly shows rate strengthening but at high velocities (440 m/yr) the relationships for the different inversions diverge: the SIA continues rate strengthening, the SSA has a kink where tractions level off, and the FS tractions decrease. Catchment 8 is almost

identical to 7 which shows rate-strengthening until ~480 m/yr where after the traction estimates diverge. However, Catchment 8 also has a small kink that occurs at ~20 m/yr.

### 3.2 *p* magnitude

The averaged flow exponent (calculated using the combined mean of SIA and SSA) within the likely frozen regions is 2.9 [2.4 – 4.1] which is expected to approximate $n$ (*Figure 6, supporting Figure S3*). In all catchments $p$ of the

thawed regions exceeds the approximate $n$ to varying degrees [4.0 – 10.6] suggesting the influence of processes that increase the nonlinearity of ice flow at the catchment scale. The highest $p$ [> 8.1] are recorded in the two northernmost and the southernmost catchment driven by a strong increase in curvature and abrupt dips in traction as velocities increase (*Figure 5*). However, for catchment 5 high $p$ and increased curvature only occur for the SSA traction estimates. $p$ of this magnitude cannot be explained through systematic variations in $n$ (*supporting information*), and

thus is consistent with the influence of basal processes (i.e. deforming till, cavitation, roughness) on catchment-scale



dynamics (Gagliardini et al., 2007; Kamb, 1991). The binned traction relationships and associated $p$ in the remaining catchments (3, 4, 6 – 8) are similar. $p$ is approximately 4.4 [4.0 – 4.9] indicating a modest increase in nonlinearity compared to the frozen regions. The changes in the deformation fraction as ice warms or thins towards the margin shown on *Figure 3* are unlikely to produce a greater than ~0.15 change in $p$ from $n$, and thus even though small, the

increase in $p$ is unlikely to be due to systematic deformation variations. This suggests a modest influence of basal processes or rheologic variations that would increase $p$ from $n$ (*supporting information*).

**3.3 Spatial distribution of strong and weak beds**

We map the distribution of weak and strong beds across the GrIS using the residual distribution from the ice sheet-averaged flow relationship derived from the mean of the three traction estimates *(Figure 7, 8)*. We classify strong and

weak beds as data that comprise respectively the top and bottom 15% of the distribution respectively, while the middle 70% of the data is classified as having normal strength (*Figure 7*). In other words, strong beds have a higher than normal traction for a given velocity, while weak beds have lower than normal traction for a given velocity, and thus the classification is independent of substrate type.  By using the mean traction, information from all three traction methods is incorporated and thus weak and strong beds will occur most frequently where the traction estimates agree

on the bed strength. We note that the variability used for this designation is far greater than that expected from data error and uncertainty (*supporting information, supporting Figure S4*) and thus represents real spatial variations in basal characteristics. We compare the spatial variability in bed strength to the distribution of fast and slow-moving ice defined as those moving greater and less than 200 m/yr respectively, as well as the decadal-averaged snowline which serves as a time-integrated proxy for the equilibrium-line altitude (Vandecrux et al., 2019).

The ice-sheet wide distributions of strong and weak regions show the majority of the ice sheet domain is of average strength with aggregated regions of strong and weak beds distributed in select locations (*Figure 8*). This general description applies for both the slow and the fast-moving regions of the ice sheet, including the three largest glaciers in Greenland: Jakobshavn Isbræ, Kangerlussuaq glacier, and Helheim glacier (Shapero et al., 2016). There are only a few visually aggregated weak-bed regions across the GrIS. The largest region occurs in catchment 2 within the bounds

of the Northeast Greenland Ice Stream (NEGIS). Aggregated weak beds also occur in catchment 6 directly downgradient from the snowline suggesting meltwater forcing is likely important to their occurrence. Aggregated strong beds occur in catchment 4 and catchment 8. In catchment 4 these primarily occur along the thawed-region boundary and could be due to the basal thermal state being misclassified or locally elevated stress coupling that arises near the transition to sliding. The strong bed regions in catchment 8 occur below the snowline near and are therefore

less likely to be misclassified or strongly coupled to frozen regions. We suggest they could instead be related to rougher topography (*Figure S6*) or possibly subglacial drainage conditions near the margins.

Catchments 3, 4, and 8 have the highest proportion of strong beds, while catchments 2 and 6 have the highest fraction of weak beds and are the only catchments with a higher proportion of weak beds than strong beds (*Figure 8*). We find an equal percentage of weak bed and strong bed (21%) grid cells in fast – flowing regions and in the slow-moving

regions the proportions are roughly equal (14 – 13%) as well. As weak beds disproportionately occur in catchment 2



(61% of all weak-bed grid cells), excluding catchment 2 produces a lower percentage of weak beds (8%) than strong beds (25%) in fast-flowing regions.

## 4. Discussion

### 4.1 Physically based traction relationships

We show that physically based velocity – traction relationships can be retrieved from the ice dynamics of Greenland. This should give confidence that our current descriptions of basal physics, where stresses are related to velocity, are appropriate to model ice motion across much of the ice sheet. We find that in most catchments the binned velocity and traction has characteristics that are expected given our understanding of basal physics and moreover, can be well represented with a generic power-law traction relationship with varying exponents at high and low velocities. This

provides a simple, catchment-specific parameterization that can be used to model the flow field and tractions at large scales (discussed in more detail in *4.4*).

Our finding of gravity-driven flow is the opposite result of an analysis by Stearns and van der Veen 2018 which found no relationship between traction and velocities for tidewater outlet glaciers on the periphery of Greenland. The two analyses are fundamentally different both in terms of methodology and scope making a straightforward comparison

difficult. The coarse grid spacing needed to infer the basal traction and the catchment-scale approach presently used mean our data-derived relationships dominantly reflect the grounded regions of the ice sheet. However, the large, partially floating ice tongues of the Humboldt (Carr et al., 2015) and Petermann (Hogg et al., 2016) glaciers in catchment 1 are resolved by our method and are found to be underlain by anomalously weak beds (*Figure 8*). This suggests tidewater regions might have mostly Mohr-Coulomb-like behavior and therefore our observations do not

directly contradict the previous findings (Stearns and van der Veen, 2018). Nevertheless, most of high-velocity regions inland from the calving face of Jakobshavn Isbræ, Kangerlussuaq glacier, and the Helheim glacier do show reasonable adherence to the derived flow relationships, which was not previously found (Stearns and van der Veen, 2018).

### 4.2 Interpretation of catchment-scale dynamics

Below we use the characteristics of the traction relationships, *p*-values, and the distribution of weak and strong bed

grid cells to interpret the bed properties and resulting dynamics in each catchment. We note that tractions for hard-bed sliding with cavitation and deforming beds have very similar characteristic features (i.e. strong increase curvature vs. kink). Here we characterize these features as Mohr-Coulomb-like behavior which could reasonably occur over both bed types and only interpret further where additional information permits. We note that our interpretations assume that the binned relationships (and thus model fits) reflect the main characteristics of the traction laws outlined in *2.4.1*

as opposed to systematic variations in bed parameters (i.e. roughness characteristics).

#### 4.2.1 Predominantly rate strengthening





The binned-velocity traction relationships in Catchments 3 and 4 have all the characteristics of Weertman-type traction laws. There is rate strengthening through the entire velocity range with no large kinks or curvature changes that would indicate Mohr-Coulomb-like behavior is induced either via cavitation or deforming till (*Figure 5*). Correspondingly, $p$ is the closest to $n$ for these catchments, indicating only a limited occurrence of higher non-linearity processes operating at the basal interface (*Figure 6*). There are relatively few high velocity low traction grid cells outside the middle 90th percent of the data suggesting that weak beds in these regions are not common (*Figure 4*) and those that do mostly occur within the slower moving regions inland from the margin, with the exception being the very terminus of the Helheim Glacier (*Figure 8*).

To a large degree the relationships in catchments 7 and 8 also adhere to Weertman-type hard bed physics behavior for all three traction estimates below velocities of ~ 450 m/yr, which accounts for greater than 95% percent of the data within each catchment (*Figure 5*). At higher velocities the traction estimates diverge and the interpretation becomes more ambiguous. The SIA continues to show rate-strengthening while the curvature of SSA and FS increase and the tractions level off. The difference between the relationships indicates that higher-order stresses and/or topographic form drag could influence the defining characteristics of the traction relationship at high velocities which is consistent with the spatially complex flow and bed topography found near the margins (Joughin et al., 2018; Morlighem et al., 2017). The magnitude of these effects is again determined by rheology, and thus the true traction field remains uncertain but could reasonably reflect either rate-strengthening or Mohr-Coulomb-like behavior given the spread in the relationships at high-velocities.

An increase in Mohr-Coulomb-like behavior at high velocities near the marine-terminating outlets would be consistent with prior traction inversions of the fastest-flowing outlet glaciers in Greenland (Jakobshavn Isbræ, Helheim, Kangerlussuaq)(Habermann et al., 2013; Joughin et al., 2012; Shapero et al., 2016). In general, these inversions have found uniformly weak beds underneath the concentrated regions of fast flow that occur within the deep bed troughs. Interestingly, these outlet glaciers also correspond to high driving stress which cannot be supported by the inferred weak beds, and thus stress is balanced by > 0.5 MPa traction bands along the margins of the bed troughs which control the flow speeds (Habermann et al., 2013; Joughin et al., 2012; Shapero et al., 2016).

While our analysis by design cannot capture the complex traction field of individual outlet glaciers, we can use our flow relationships to estimate flow speeds expected at tractions of 0.5 MPa and provide a test of the weak-bed hypothesis in these regions using our data-constrained flow relationships. We find that flow speeds at 0.5 MPa would range from ~ 20 km/yr to 100 km/yr (spread from the rate-strengthening catchments). This exceeds the ~ 5 - 10 km/yr velocities that are observed in these regions which would better correspond with tractions ranging from ~0.275 - 0.4 MPa. This suggests short-length scale bed roughness where the high stresses occur may be exceptionally rough and support higher stresses than expected for the given flow speeds, or alternatively the ice in the shear margins is weaker and produce less resistance than expected, and thus beds within the trough may still be weak, but not as weak, or as uniformly weak as previously inferred.

### 4.2.2 Predominantly Mohr-Coulomb-like





Catchment 1 and 2 both show strong evidence of Mohr-Coulomb-like behavior. In both catchments the average $p$ is greater than > 8.1 (*Figure 6*), driven by abrupt increases in curvature in the binned velocity - traction relationship which is characteristic of Mohr-Coulomb beds (*Figure 4, 5*). Further, the dip in the binned relationship in Catchment

2 indicates that weak beds make up a substantial fraction of grid cells at high velocities.

The distribution of grid cells indicates the beds in each catchment have different characteristics even though they both display Mohr-Coulomb-like traction relationships. In catchment 1, the change in curvature coincides with an increase of high velocity – low traction grid cells (*Figure 4*) which occur within the partially floating ice tongues of the Humboldt and Petermann glaciers (*Figure 8*), indicating that weak beds mainly coincide with areas where the

influence of sea-level on basal water pressures can occur (Carr et al., 2015; Hogg et al., 2016). In catchment 2 low traction grid cells occur throughout the entire velocity range and become more dominant at higher velocities (*Figure 4*). Unsurprisingly, they closely coincide with the NEGIS (*Figure 8*) which is known to be underlain by mechanically weak till (Christianson et al., 2014) and has high rates of basal melting from an anomalous geothermal heat flux (Fahnestock et al., 2001), factors which are considered critical to its occurrence (Christianson et al., 2014; Fahnestock

et al., 2001; Rogozhina et al., 2016).

### 4.2.3 Mixed behavior

The binned relationship for catchment 6 is predominantly rate strengthening and has a good fit to the power law models which only a show small increase in $p$ over $n$ (*Figure 4 – 6*). Yet the catchment also has the second highest percentage of weak-bed grid cells (*Figure 8*) that sharply increase in occurrence at 60% through the velocity range

which coincides with a small kink in the traction relationship (*Figure 4, 5*). Correspondingly, the delineation of the bottom 5% of the data (black markers, *Figure 4*) in each bin also shows a sharp Mohr-Coulomb-like kink. Thus, while there are not enough weak-bed grid cells to significantly impact the median of the binned relationships, weak beds are clearly present and important to the ice dynamics in this catchment. This characterization is consistent with direct measurements of basal conditions that have identified both hard and till beds (Booth et al., 2012; Dow et al., 2013;

Harper et al., 2017; Kulessa et al., 2017) and persistently low effective pressures that are almost always less than 20% of overburden (Wright et al., 2016).

The weak-bed grid cells mostly occur directly downgradient from the snowline and presumably are linked to the occurrence of hydrologic forcing (*Figure 8*) which matches previous conclusions of a previous study which identified the weak bed regions as occurring due to the onset of hydrologic forcing (Meierbachtol et al., 2016). In these higher

elevation zones, moulins are sparsely spaced (Smith et al., 2015), but large, stochastic, injections of water to the base from supraglacial lake drainages are thought to weaken the bed during the melt season (Das et al., 2008; Kulessa et al., 2017; Stevens et al., 2015). Interestingly, this region is the only region to show a transition to weak beds below the snowline even though supraglacial lakes and meltwater forcing are ubiquitous along the margins of the ice sheet (Leeson et al., 2015).

**4.2.4 Unclassified**



There is no clear interpretation for the relationship in the catchment 5 which has the fewest grid cells (*Figure 1*) and the highest ice thickness and velocity errors that are often near 50% magnitude (*supporting Figure S5*). Unlike the other catchments, $p$ values for the southernmost catchment are notably different for the SIA ($p = 4.4$) and SSA ($p = 16.9$) (*Figure 4*). While there is a well-defined kink in the flow relationship, after the kink the SIA and FS again show
rate strengthening while the SSA shows Mohr-Coulomb behavior (*Figure 5*). It is generally unclear why the FS has higher tractions, which due to the inclusion of topographic form drag should have lower tractions than the SSA. The kink occurs at low velocity and high stress inland from the ELA (*Figure 8*), which could reasonably be a result of misclassification of basal thermal state as thawed or a transition to a Mohr-Coulomb-like bed. Given these uncertainties, we refrain from characterizing the catchment in further detail.

**4.3 Grounded regions of the ice sheet mainly set by hard-bed physics**

The ice-sheet wide velocity-traction relationship exhibits rate-strengthening which is typical of hard-bed physics (*Figure 7*). This is consistent with the catchment scale results where most of the ice sheet outside of catchment 1 and 2 are generally consistent with a Weertman-type traction law, even regions with ice speeds approaching 1000 m/yr. Our results dominantly reflect grounded regions of the ice sheet where sea-level does not set the reference pressure,
yet even so, our results suggest that even near terminus of the largest marine-terminating outlets glaciers, hard-bed physics primarily dictates the traction field. The $p$-values in the rate-strengthening catchments are greater than expected due to regelation alone ($p \approx \frac{n+2}{1}$)(Weertman, 1964), which indicates the main rate-strengthening process is likely form drag around small-scale bed roughness.

We cannot explicitly discriminate between bedrock beds and non-deforming till beds which are expected to have
identical sliding physics in the rate-strengthening range (Gagliardini et al., 2007; Weertman, 1964; Zoet and Iverson, 2020). However, comparison to direct observations made via seismic and borehole survey indicate till beds are relatively common across the western half of Greenland in regions where we identify the bed to predominantly reflect Weertman-type sliding (Booth et al., 2012; Dow et al., 2013; Doyle et al., 2018; Kulessa et al., 2017; Walter et al., 2014). Interestingly, these observations contradict radar-derived roughness proxies at the ice-sheet scale that find
significant bed roughness at the < 100 m (Cooper et al., 2019) and km length scale (Lindbäck and Pettersson, 2015; Rippin, 2013) which is inconsistent with flat beds typically associated with sediment plains. Thus, there is a discrepancy between the local and ice-sheet wide indicators of substrate and our observations of rate-strengthening.

To reconcile the observations and the results of our study, we posit that even though till beds appear to be relatively common across the grounded regions of the ice sheet, they are either weak and predominantly occur in small patches,
more widespread but mechanically strong, or not always deforming in time. As a result, even across mixed-substrate domains, hard-bed physics is manifested in the ice flow field and surface geometry at large scales in Greenland. Our assertion of either dispersed and patchy or mechanically strong till beds is supported by the stress regimes coinciding with the direct till observations made on the western margin of Greenland. The five field sites where till was directly identified indicate overall high bed strength with driving stresses between 0.08 and 0.21 MPa (Booth et al., 2012;



Doyle et al., 2018; Kulessa et al., 2017; Ryser et al., 2014; Walter et al., 2014) (*supporting Table S2*). Further, the occurrence of multiple observations of hard-bed conditions (Harper et al., 2017) in the same region as many of the till observations (Booth et al., 2012; Dow et al., 2013; Kulessa et al., 2017) suggest that till naturally occurs in discontinuous patches along the bed which is consistent with our interpretation.

**4.4 Implications for Greenland ice flow**

Our comparison of multi-year velocities and tractions shows the adherence to rate strengthening averaged across summer and winter flow which suggests over longer timescales much the dynamics at the ice-sheet scale are largely determined by the stress at the bed. Future changes in ice speeds will be dictated by marginal thinning, changes at marine-terminating outlets, or changes in the basal hydrology (Alley and Joughin, 2012).  Our flow relationships are likely not prognostic at the seasonal to annual time scales where changes in the gravitational stresses are small and

flow variations predominantly reflect changes in the subglacial drainage system or changes at marine terminating outlets. Yet, where there are large changes in the traction due to thinning, our flow relationships can estimate the corresponding change in dynamics. Using our model parameters from the rate-strengthening catchments we estimate the rate at which velocities will change in response to traction perturbations. In fast-flowing regions (~1000 m/yr) the rate of  velocity change is estimated between 15 and 35 m/yr per kPa and in slow regions (~200 m/yr)  between 4 and

10 m/yr per kPa (*supporting Figure S7, Table S1*). This serves as an important constraint in Greenland where much of the marginal zones are observed to be thinning (Csatho et al., 2014; Mouginot et al., 2019).

Across timescales and in regions where meltwater forcing drives flow variations, one of the most critical characteristics that determines how ice flow will respond is whether the bed is hard bedrock or deformable sediment. This property not only sets the type of physics at the bed, but also sets the morphology of the basal drainage system,

where water either flows at the basal interface through networks of cavities and channels or conversely through the pore space of the till (Cuffey and Paterson, 2010).  While our analysis cannot explicitly distinguish Mohr-Coulomb-like behavior induced by till deformation versus extensive cavitation, we posit that the weak-bed grid cells are reasonable candidates for where there could be consistently deforming till beds. Modeling has suggested deforming till beds are much more likely to experience continued acceleration with increasing melt (Bougamont et al., 2014),

and correspondingly, these regions could be more vulnerable to increased meltwater forcing in a warming climate. Interestingly, there are very few weak-bed grid cells found below the snowline where active meltwater forcing occurs except in catchment 6. This is stark contrast with catchments 7 and 8, where bed strength below the snowline is mainly normal or strong. This may indicate hydrology has a limited effect on multiyear averaged ice flow outside of select regions, but importantly, also reinforces the idea that till beds in other regions outside of catchment 1 and 2 are either

mechanically strong or only sporadically undergo periods of deformation likely during periods of active melt forcing.

**5. Conclusions**

We use multi-year averaged velocities and basal traction estimates to determine the spatial relationship between velocity-traction relationship for all eight drainage catchments in Greenland.  We show the derived traction



relationships are consistent with our current understanding of hard and deforming bed physics and that we can clearly

distinguish between predominantly rate-weakening and predominantly Mohr-Coulomb-like behavior in different

catchments. Overall, we find the flow field and ice geometry of Greenland mostly reflect Weertman-type sliding

physics across the grounded regions of the ice sheet, with the exception being the NEGIS which obeys expectations

given Mohr-Coulomb-like behavior. We use the ice-sheet wide flow relationship to identify the location of weak beds

which constrains the regions that may have deforming beds that are more vulnerable to increased melt in a warming

climate. Overall, our work indicates that dynamics over grounded regions of Greenland will largely depend on

Weertman-type hard-bed sliding physics which provides a simple framework for modelling current and future ice

flow.




**Figures**


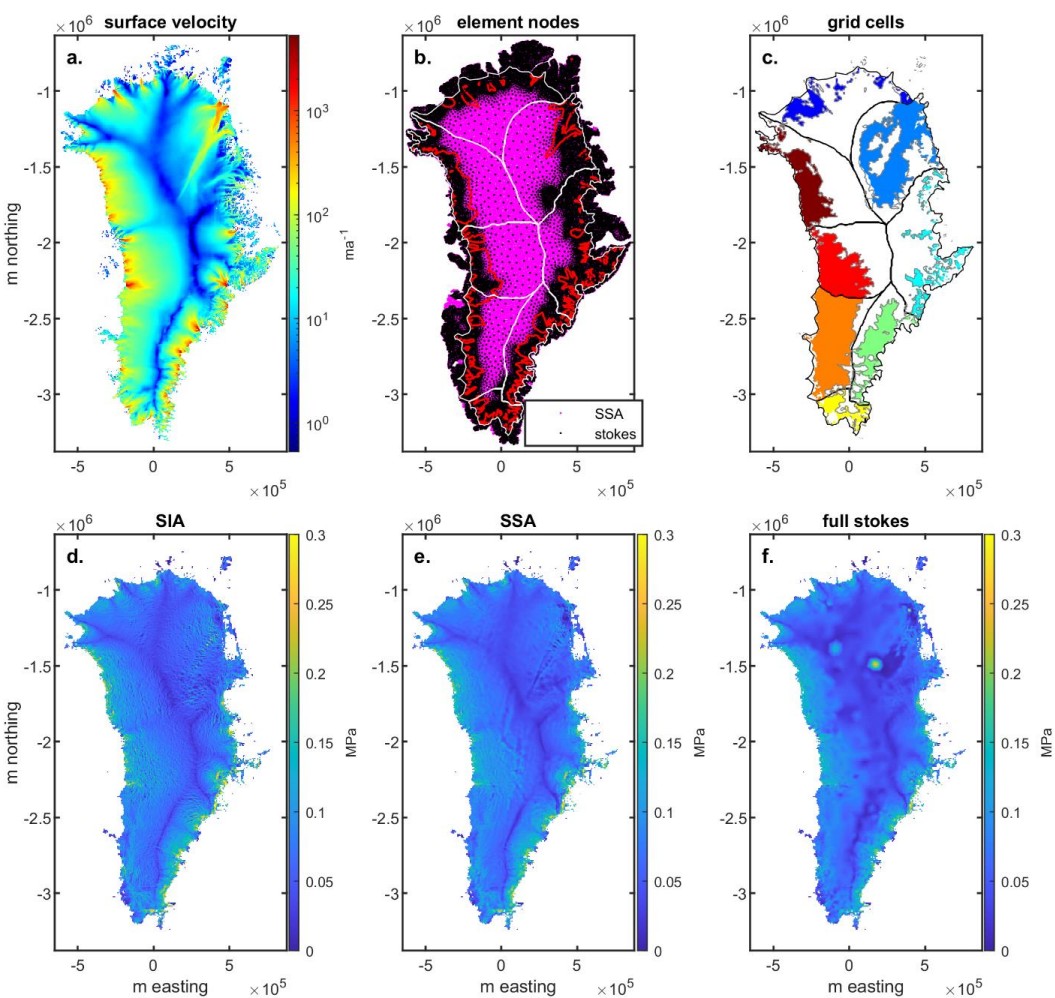

**Figure 1: a.** Multi-year surface velocity mosaic for GrIS (Joughin et al., 2018). **b.** Element nodes for the SSA (pink dots) and FS (black dots) inversions. The red contour shows the 100 m/yr velocity contour which delineates the regions from the stokes inversion used in the analysis **c.** Map of regions used in analysis. Black lines show drainage catchment delineation. The grey line indicates boundary of likely-thawed region. **d.** Driving stress calculated over 6 km x 6 km grid cells. **e.** Basal traction from SSA inversion averaged over 6 km x 6 km grid cells. **f.** Basal traction from FS inversion averaged over 6 km x 6 km grid cells.



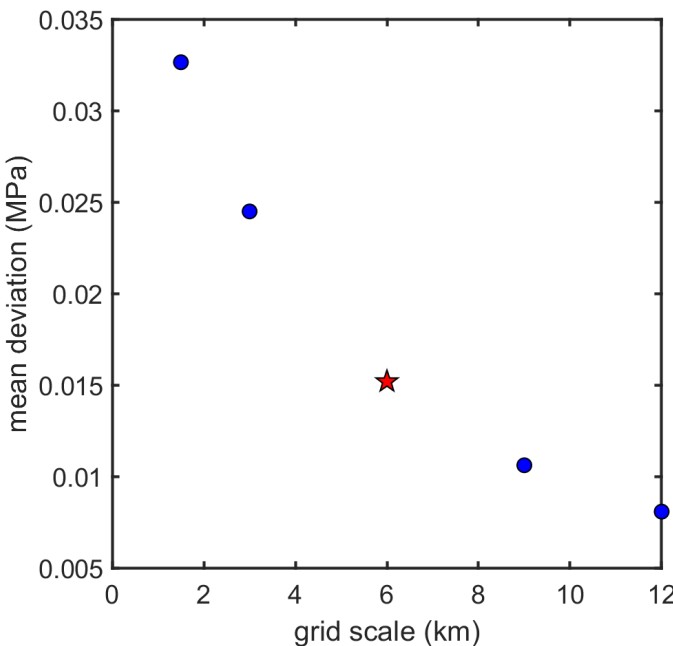

**Figure 2:** Mean deviation ( magnitude of SIA minus SSA) between the SIA and SSA approximation of basal traction
as a function of grid scale. Decreasing deviation indicates the effect of higher-order stresses decreases with increasing
grid length. Red star shows 6 km grid scale which is the primary grid scale used for our analysis.


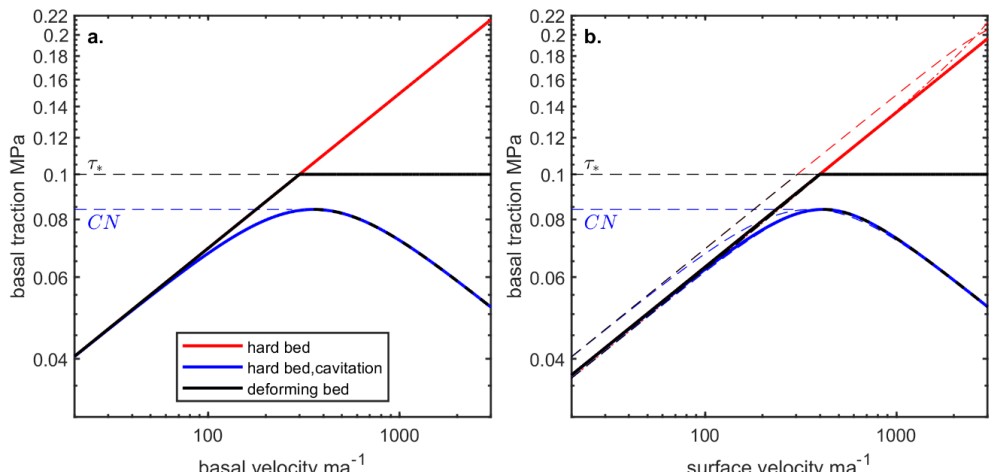

**Figure 3: a.** Traction models that relate basal motion to basal traction for hard beds with and without cavitation and for deforming beds. $CN$ indicates Iken's bound and $\tau_*$ is the critical shear stress for deforming till. **b.** The same relationships are shown with added deformation motion calculated with $T = 0°$ and $H = 2500$ m. The bounds show deformation motion where parameters in the deformation calculation change with velocity. For the higher bound $H$ is held constant at 2500 m while $T$ increases from -25° to 0° through the velocity range. The lower bound, $T$ is held constant at 0° and $H$ decreases from 3000 m to 200 m through the velocity range.







**Figure 4:** The velocity – traction relationship for the SIA, SSA, and Full Stokes inversions for all eight drainage catchments. Transparency is an indicator of data density, where opaque areas indicate four or more grid cells occupy the same marker area on the plot. Magenta markers show the median traction values for 20 logarithimically spaced velocity bins, the vertical bars show the interquartile range, and the black dots indicate the middle 90 percent of the





data. Black line shows a power law model fit to the binned data. No models were fit to the stokes relationships due to

the velocity range (> 100 m/yr) which is limited in our analysis.


**Figure 5:** The relationship between velocity and traction for the SIA (solid line), SSA (dashed line), and FS (dot-dash line) are shown together for each catchment. The right axis shows the cumulative distribution function of the data (grey line) moving through the velocity range.

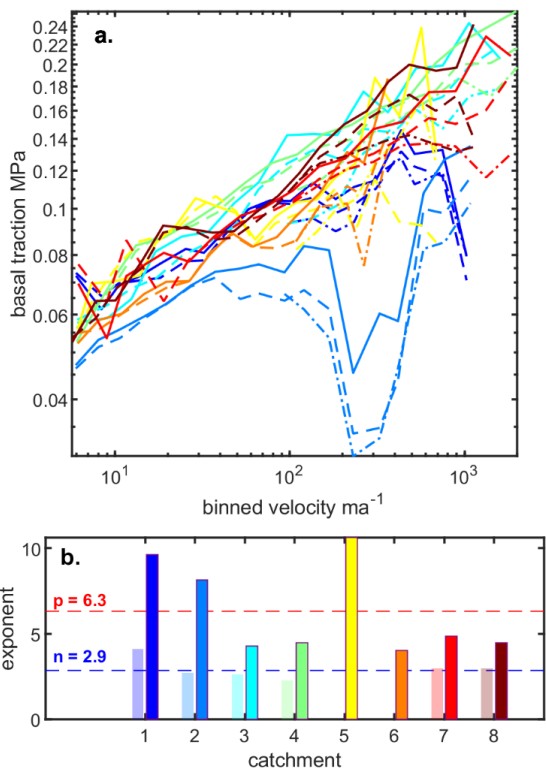

**Figure 6: a.** Binned velocity versus traction for all catchments and traction estimates (SIA – solid, SSA – dashed, FS – dash-dot). **b.** $p$ for likely thawed regions (dark bars) and $n$ for likey frozen regions (light bars) in each catchment. Values for each catchment are the average from the SIA and SSA. Dashed lines show the mean value for each exponent. No $n$ is shown for catchment 5 and 6 which did not have enough grid cells to produce a reliable relationship (*supporting Figure S3*).

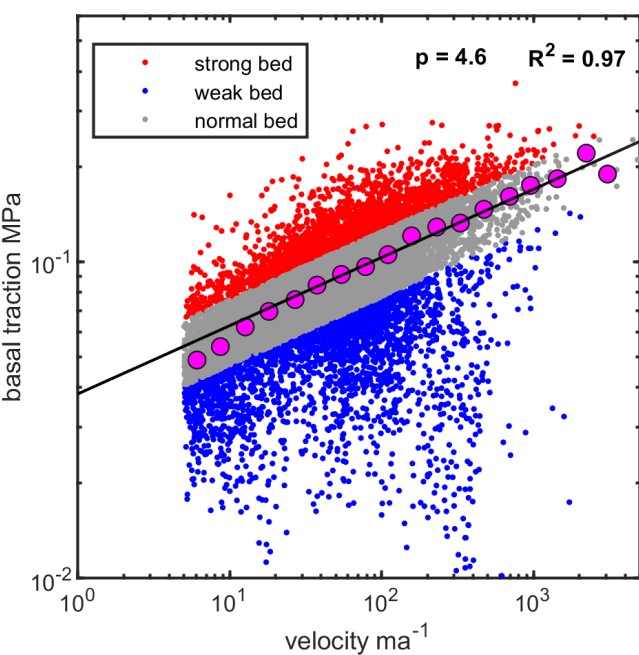

**Figure 7:** Weak and strong bed delineation is presented based on residuals from ice sheet-wide flow relationship. Ice
sheet-wide flow relationship (black line) is calculated by averaging the SIA, SSA, and FS traction estimates for each
grid cell in the likley thawed regions and fitting a power law model to the binned data. Pink markers show the binned
velocity – traction relationship (median, 20 bins, logarithmically spaced). The residuals from the model relationship
are used to delineate and map bed strength throughout Greenland. Strong and weak beds comprise the bottom and top
15% percent of the log-residual distribution, while grid cells with normal bed strength occupies the middle 70%.
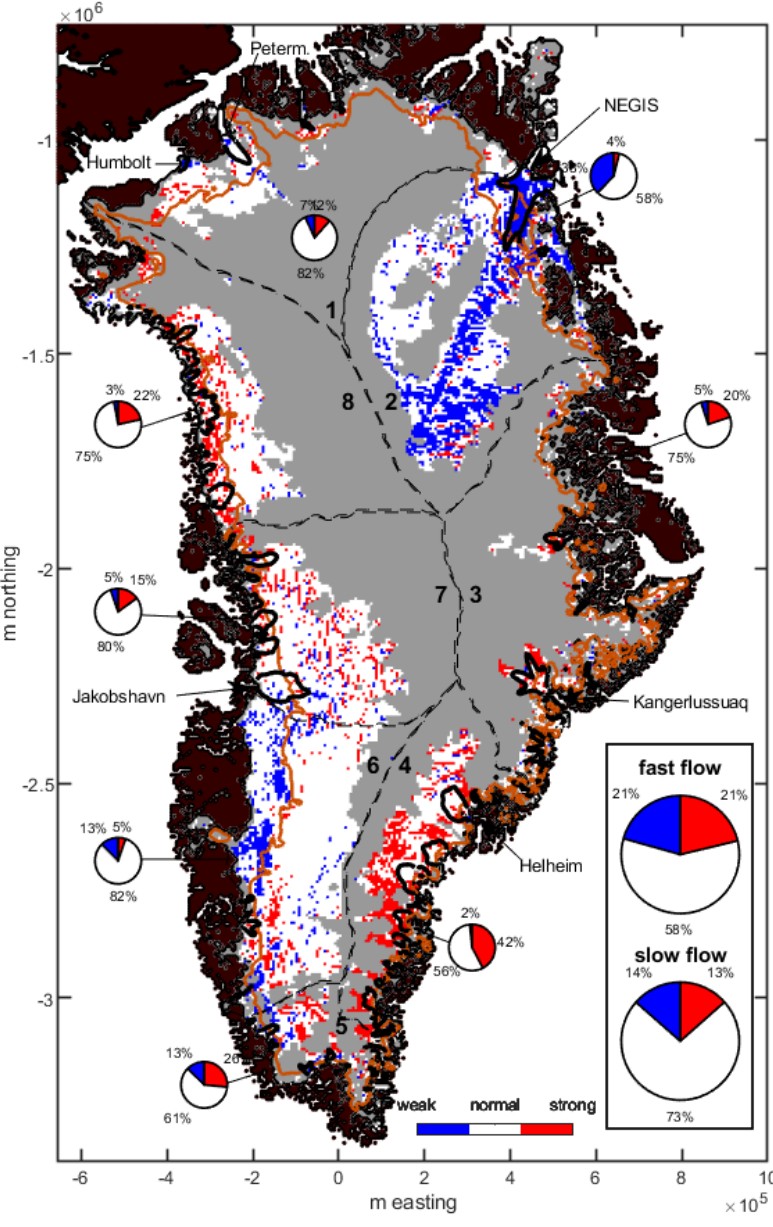

**Figure 8:** Map of weak and strong bed grid cells. Black contours separate fast and slow flowing regions of the ice sheet (200 m/yr contour). Orange line delineates the average snowline from 2001 to 2017 (Vandecrux et al., 2019). Dashed lines delineate flow catchments. The small pie charts show the percentage of weak and strong regions within each catchment. The large pie charts indicate the percentage of weak and strong regions within the slow and fast moving regions.



**Data and Code Availability:** All data used to generate this manuscript is currently archived and publicly available for download. Multi-year surface velocities: doi.org/10.5067/QUA5Q9SVMSJG  Ice Surface Elevation: doi.org/10.5067/H0KUYVF53Q8M, Land/Ice classification: doi.org/10.5067/B8X58MQBFUPA, Bed topography: doi.org/10.5067/2CIX82HUV88Y, Basal thermal state map: doi.org/10.5067/R4MWDWWUWQF9, Snowline
elevation: doi:10.18739/A2V40JZ6C. Elmer/Ice code/output is available upon request.

**Acknowledgments:** This work was funded by the French National Research Agency grant ANR-17-CE01-0008.

**Author Contributions:** N.M. designed the study, analyzed the data, produced the figures, and is the primary author of the manuscript. F.G. contributed significantly to the data analysis and writing of the manuscript. F.G.-C. performed the basal traction inversions that were implemented with Elmer/ice and contributed to the writing of the
manuscript. A.G. contributed to writing of the manuscript.

**Competing Interests:** The authors declare no competing interests.

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
