# Peer review of "Basal traction mainly dictated by hard-bed physics over grounded regions of Greenland"

_The Cryosphere, 2020_

## Referee Comment (RC1) · Bradley Lipovsky (Referee) · 13 Oct 2020

Dear Maier and coauthors,

I am happy to have been asked to review this paper. Its topic and methods are at the cutting edge and the results will be interesting to readers of the Cryosphere. I'd like to highlight several ways that the paper could be improved.

Sincerely,

Brad Lipovsky

1. Length scale for grid cells. I like the approach in Section 2.4 but I think it could be made better:

[Figure]

- My main concern is that Figure 2 compares different regions with different force balance approximations. Wouldn't it be better to compare the *same* regions with different force balance approximations? In other words, to carry out a grid-refinement study, and then examine the convergence behavior of different force balance approximations? -I would expect the relevant length scale for SIA-SSA differences to be quite high in the interior (far greater than 6km) but much smaller near the margins (perhaps even less then 6km). - Figure 2 could be on a log scale so we can see if there is a change in the power law. - The 6km cutoff seems arbitrary. I don't see anything special about 6km in Figure 2. Again, if you plot this on log-log axes I have a feeling it will make a straight line (i.e., a power law) with nothing special about the 6km cutoff.

2. How well resolved is the sliding law at high velocity? Of course there are many fewer data points in this regime because most of the ice sheet is slow-flowing. One idea is that, in Figure 5, plotting PDFs rather than CDFs would more accurately convey to the reader that there are very few data points at high velocity. This is an important point to convey.

Concerning Catchment 2, the increase in basal traction at high velocities is very strange. I'm worried that this increase is based on only a very small number of data points (CDF is almost flat at high velocities). Question: Is there some way that you could re-draw Figures 5 to *combine* the CDF and velocity/traction curve? Example: what if the opacity of the curve was set by the number of observations? Drawing the curves in this way would highlight the areas that are more well captured by the data. This is just a guess about how to convey this point. . . no worries if it doesn't work out.

3. Many catchments show rate strengthening at high velocities, a non-intuitive result for me. Based on Figure 3, I would have thought that these velocities were high enough to reach rate-weakening behavior. From Gagliardini's paper, Equation 24, we may therefore place bounds on subglacial parameters. It would be interesting to see if you could constrain A or C from your observations.

---

## Referee Comment (RC2) · Anonymous Referee #2 · 20 Oct 2020

This is a well-written manuscript that reflects a lot of hard work by the authors. It sets up an interesting and thoughtful experiment, assessing basal traction with three different model hierarchies.

Main Concerns: - A large basis of the conclusion relies on the relationships in the scatter plots in Figure 4. These are on a log-log scale and even then the scatter is so great that I am skeptical any credible conclusion can be drawn from these graphs. Binning the velocities and taking the median, especially in log-log space looks better, but does seem kind of sneaky. I am particularly nervous about the unequal distribution of velocities, since fewer data points exist in the faster flowing regions in making the bins. So, while 'fitting the binned data gives equal weights to fast flowing and slow-flowing regions of each catchment" (line 247), many of the faster flowing bins only

have a handful of data points, so the fit is biased by the slower regions.

I am sympathetic that these 'tricks' are needed in order to get the clean-looking relationships that are distilled in Figure 5. However, Figure 4 really highlights the incredible complexity of this relationship that gets glossed over in the subsequent discussion and conclusions. Instead, every kink in the fitted curve is analyzed in great detail. Figure 8, which shows a map of strong vs weak bed grid cells, takes this (in my opinion) messy relationship and derives some binary conclusions about the preponderance of hard-bedded rheology. At 6 km data points, then binned to derive the traction-velocity relationship, the impact of the fastest flowing regions is muted.

- The assumption (section 2.2.1) that basal drag can be approximated by the driving stress works for the interior but really falls apart near the margins. As a result, I think the limit of this analysis precludes a lot of the faster flowing regions. Given that shortcoming, the overall conclusions are weaker.

- While it is done frequently, it is incredibly circular to prescribe a sliding relation to estimate basal drag and then use the calculated basal drag to show a correlation with velocity (section 2.2.2 and 2.2.3). The whole justification for this is described in line 137: While a linear sliding law is used for the inversion, the traction field is shown to have little sensitivity to the choice of sliding law (Joughin et al., 2004)." This paper uses the control method exclusively for the Ross Ice Streams (no basal drag and high velocity), which is opposite to how it's being used here.

Other comments:

Line 10: "three different methods" but with the same assumptions and workflow. It would be clearer to say models with three different levels of complexity. This comes up again in line 83ish. Having three models that are based on the same assumptions converge does not necessarily yield confidence in the velocity-traction relationship – just highlights where the complexity is unimportant.

Line 46: "bypassing" is too strong a word. We have some models that can be used to infer basal processes, but they are really far from capturing details from localized observations at the ice bed interface.

Line 76: How can rheology have impacted the findings of Stearns and van der Veen? This is a fairly bold statement; if you keep it in it needs justification. I think it's ok to say it's still under debate and leave it there.

Line 254: What is meant by "visually well defined" and "show good agreement with the model fit at both high and low velocities". I thought the model was fit to these binned points.

Line 255: High R2 (.22 -.99)? It's more representative to specify where it's high and where it isn't - especially since it's closer to .99 in most places. Also specify what the R2 value refers to (binned or raw data).

Figure 5: Add labels of catchment to each subplot for color-blind folks. Is the grey line the cdf of the binned velocity data or the input data? I'm guessing the latter, which is misleading because the velocity-traction curves are based on the binned data. Don't these graphs also show that the models diverge at fast flow (I didn't like framing the models this way at the start, but since you did it tshould be revisited).

Figure 8: Doesn't the fact that strong and weak beds exist roughly equally in slow and fast flow regions show that bed strength is not related to velocity? If so, then a sliding relation that equates the two falls apart.

---

## Author Comment (AC1) · 13 Nov 2020

Thank you for the detailed comments which helped improve our manuscript. We have attached our responses to your individual comments in the attached supplement.

Please also note the supplement to this comment:
https://tc.copernicus.org/preprints/tc-2020-185/tc-2020-185-AC1-supplement.pdf
* * *

---

## Author Comment (AC2) · 13 Nov 2020

**Response to Reviewer Comments for:**

**"Basal traction mainly dictated by hard-bed physics over grounded regions of Greenland"**

**Comments from Bradley Lipovsky (Referee)**

**Comment:** Dear Maier and coauthors,

I am happy to have been asked to review this paper. Its topic and methods are at the cutting edge and the results will be interesting to readers of the Cryosphere. I'd like to highlight several ways that the paper could be improved.

Sincerely,

Brad Lipovsky

**Response:** Thank you for taking the time to review the manuscript and making very constructive comments.

**Comment:** 1. Length scale for grid cells. I like the approach in Section 2.4 but I think it could be made better: My main concern is that Figure 2 compares different regions with different force balance approximations. Wouldn't it be better to compare the *same* regions with different force balance approximations? In other words, to carry out a grid-refinement study, and then examine the convergence behavior of different force balance approximations? -I would expect the relevant length scale for SIA-SSA differences to be quite high in the interior (far greater than 6km) but much smaller near the margins (perhaps even less then 6km). - Figure 2 could be on a log scale so we can see if there is a change in the power law. The 6km cutoff seems arbitrary. I don't see anything special about 6km in Figure 2. Again, if you plot this on log-log axes I have a feeling it will make a straight line (i.e., a power law) with nothing special about the 6km cutoff.

**Response:** Thank you for the detailed comment. We are assuming this comment addresses Section 2.3 rather than 2.4 since it is the section addressing length scales. We shall clarify first that in the analysis presented in Figure 2 the mean difference between the data SIA and SSA data *is* derived from directly comparing the traction for the same grid cells across the ice sheet (Fig. 2) which is outlined in Section 2.3. However, as you highlight, our length-scale justification is a compromise between the ideal physically based choice and more practical considerations. In the ideal case, the length scale would be optimized when the difference between the SIA and SSA no longer decreases moving up to a coarser grid cell implying the influence of stress gradients are perfectly minimized. However, our length scale choice also considers several other criteria that we decide are needed to fit the requirements of our study. Specifically, we take into consideration the resolution near edges of the ice sheet and the span of the velocity range used to derive the relationships. Each of these requirements favor shorter length scales. In the end, we settle on 6km because it minimizes the difference between the SSA and SIA reasonably well (Figure. 2), and it maximizes the resolution and inclusion of grid cells near the margin as well as the velocity range. Most importantly we show that moving up to a larger grid cell does not change our interpretation. This rational for this decision is outlined in lines 159-163 in the main text of the original manuscript.

In terms of the technical aspects of the length scale optimization, the reviewer correctly points out that the longitudinal coupling length scale varies with ice thickness. Thus, the ideal length scale where stress gradients are minimized would vary across the ice sheet and would be significantly different for thick and thin ice. To simplify our analysis, we assume that at the chosen length scale stress gradients are minimized reasonably well everywhere. This includes the thinner margin regions where ice flow is more complex, as well as the thick interior where flow varies smoothly and slowly in space. To show that the 6 km length scale is a reasonable choice across a wide range of ice thicknesses and settings, we separate the data set into thick and thin ice (below and above 1500 m ice thickness), and inland and interior regions (below and above 1200 m.a.s.l.) to see how much the difference between the SIA and SSA changes. We find at 6 km there only a small change (0.004 MPa) in the difference between the SSA and SIA between the four different data subsets (Figure R1). While it may seem counter intuitive for the difference to be less for the interior grid cells where the longitudinal coupling distance (~10 x ice thickness) are expected to be greater, ice flow is generally much less complex than near the margins, and thus magnitude of the stress gradients in general are likely lower.

We also address the reviewer suggestion of having Figure 2 of main text shown in log-log (Figure R2). In log space the optimization does not show true power law behavior and still has curvature. We feel that for the linear scale plot is more intuitive for the reader and keep the linear scale plot in the main text.

[Figure]

**Figure R1:** Mean traction difference (magnitude of SIA minus SSA) between the SIA and SSA approximation of basal traction as a function of grid scale. Decreasing deviation indicates the effect of higher-order stresses decreases with increasing grid length. Red star shows 6 km grid scale which is the primary grid scale used for our analysis. Dashed lines show same analysis for different subsets of the data (margin regions < 1200 m.a.s.l. – yellow line, interior regions > 1200 m.a.s.l. – grey line, thickness > 1500 m – magenta line, thickness < 1500 m – cyan line).

[Figure]

**Figure R2:** Same figure as above but in log space. Mean traction difference (magnitude of SIA minus SSA) between the SIA and SSA approximation of basal traction as a function of grid scale. Decreasing deviation indicates the effect of higher-order stresses decreases with increasing grid length. Red star shows 6 km grid scale which is the primary grid scale used for our analysis. Dashed lines show same analysis for different subsets of the data (margin regions < 1200 m.a.s.l – yellow line, interior regions > 1200 m.a.s.l – grey line, thickness > 1500 m – magenta line, thickness < 1500 m – cyan line).

**Comment:** 2. How well resolved is the sliding law at high velocity? Of course there are many fewer data points in this regime because most of the ice sheet is slow-flowing. One idea is that, in Figure 5, plotting PDFs rather than CDFs would more accurately convey to the reader that there are very few data points at high velocity. This is an important point to convey. Concerning Catchment 2, the increase in basal traction at high velocities is very strange. I'm worried that this increase is based on only a very small number of data points (CDF is almost flat at high velocities). Question: Is there some way that you could re-draw Figures 5 to *combine* the CDF and velocity/traction curve? Example: what if the opacity of the curve was set by the number of observations? Drawing the curves in this way would highlight the areas that are more well captured by the data. This is just a guess about how to convey this point. . . no worries if it doesn't work out.

**Response:** While we give equal weighting to the high and low velocity regions through median binning, some of the bins have much less data, and as you point out, there is less certainty in the central tendency at high velocities. However, since each relationship is catchment specific, the median of the high velocity bins still does reflect the data in the catchment.  To make it clear that there is less data at high velocities, we add a PDF as the reviewer suggested in Fig 5.  We also tried different variants of the opacity idea you proposed, but found it makes the figure a more confusing.

Concerning catchment 2, we do find increasing traction at high velocities which indicates that in fastest flowing regions of this catchment we likely have grid cells that are probably more consistent with rate

strengthening. The traction relationships (bins, fits, etc.) presented in the manuscript are for large areas where spatially varying bed properties are likely. Accordingly, the traction relationships are meant to be interpreted in terms of the predominant physics that would produce the main features relationship in each catchment (i.e. rate strengthening or Mohr-Coulomb-like behavior), rather than taken at face value. As you point out, we have some catchments that show "mixed" behavior, and thus our relationships do not perfectly match the idealized traction relationships presented in Figure 3. However, in the case of catchment 2, there are very few data points that show high velocities and high tractions, and thus the bulk catchment behavior is best characterized as Mohr-Coulomb-like as outline in Section 4.2.2. To help clear up any confusion on how we draw conclusions on the predominate behavior in each catchment, we added additional text at the beginning of section 4.2 (lines 355-357).

**Comment:** 3. Many catchments show rate strengthening at high velocities, a non-intuitive result for me. Based on Figure 3, I would have thought that these velocities were high enough to reach rate-weakening behavior. From Gagliardini's paper, Equation 24, we may therefore place bounds on subglacial parameters. It would be interesting to see if you could constrain A or C from your observations.

**Response:** The adherence to rate strengthening for ice speeds near 1000 m/yr in some catchments was surprising for us as well and interestingly demonstrates in catchments 3 and 4 of the ice sheet, very high velocities can simply arise from high stress. This also implies that $CN$ for these catchments is also high (discussed further below). Figure 3 was not meant for quantitative comparison, but instead was meant to be demonstrative of the qualitative features that can be used to distinguish the three basic types of traction relationships (i.e. the kink and increased curvature expected for deforming beds and hard beds with cavitation). As such, the value of $CN$ which sets Iken's bound for the Gagliardini curve (black curve, *Fig. 3*) is set arbitrarily to fit the bound in the in the plotting space. We clarify this in the figure caption to avoid further confusion.

Constraining physical parameters using our traction relationships is definitely a logical idea. However, we hesitate to do so in all cases because depending on the parameter of interest, differing degrees of assumptions and simplifications are needed. The big distinction between our relationships and sliding relationships is that the ice motion used to derive them includes deformation which would influence some of the hard bed sliding parameters if calculated directly from the relationship. As per the traction law presented in Gagliardini et al., 2007 (also defined as *Eq. 3* in our paper), $C$ is not independent from $N$. Yet $CN$ can still be constrained as a lumped parameter. Since $CN$ is a stress it can be confidently inferred using a flow relationship that includes deformation. We provide a lower bound on $CN$ for Catchment 3 and 4 which are the catchments that clearly show rate strengthening through the entire velocity range for all the inversions indicating hard- bed behavior is likely. We add text to section 4.2.1. to explicitly quantify this in the manuscript (Lines 371-374).

In the Mohr-Coulomb-like catchments we find increases in curvature at high velocities which could be consistent with either Iken's bound $CN$ or the till strength, $\tau_*$. (Fig. 3-5). We add text to section 4.2.2. to explicitly quantify this in the manuscript (Lines 405-407).

$A_s$ can be also constrained using the traction law given in Gagliardini et al. 2007 from the rate parameter $C_p$ and exponent $p$ of *Eq. 3* by assuming bed shape parameter $q = 1$, $p = n$ ($n$ is the nonlinear flow exponent of ice), $CN$ is large, and the surface velocity is roughly equivalent to the sliding velocity. In this case the relationship will collapse back into a Weertman relationship and thus can be approximated using

*Eq. 2.* Given these assumptions we find $A_s = 1.1 \cdot 10^6$ and $4.4 \cdot 10^5$ m yr$^{-1}$ MPa$^{-m}$ for the SSA and SIA respectively in catchment 3, and $A_s = 3.2 \cdot 10^6$ and $6.7 \cdot 10^5$ m yr$^{-1}$ MPa$^{-m}$ in catchment 4. The large range reflects high sensitivity to small differences in $m$ found in the different inversions. Given the uncertainty in $m$ (presented in Table S1), the assumptions needed to simplify the traction relationship ($q$ is unlikely to equal 1), and importantly that $C_p$ was derived using surface velocities, we choose not constrain the sliding parameter $A_s$ in the manuscript.

**Comment:** This is a well-written manuscript that reflects a lot of hard work by the authors. It sets up an interesting and thoughtful experiment, assessing basal traction with three different model hierarchies.

**Response:** Thank you for taking the time to review it and providing us with thoughtful and constructive criticism.

**Comment:** Main Concerns:

- A large basis of the conclusion relies on the relationships in the scatter plots in Figure 4. These are on a log-log scale and even then the scatter is so great that I am skeptical any credible conclusion can be drawn from these graphs. Binning the velocities and taking the median, especially in log-log space looks better, but does seem kind of sneaky. I am particularly nervous about the unequal distribution of velocities, since fewer data points exist in the faster flowing regions in making the bins. So, while 'fitting the binned data gives equal weights to fast flowing and slow flowing regions of each catchment" (line 247), many of the faster flowing bins only have a handful of data points, so the fit is biased by the slower regions.

I am sympathetic that these 'tricks' are needed in order to get the clean-looking relationships that are distilled in Figure 5. However, Figure 4 really highlights the incredible complexity of this relationship that gets glossed over in the subsequent discussion and conclusions. Instead, every kink in the fitted curve is analyzed in great detail. Figure 8, which shows a map of strong vs weak bed grid cells, takes this (in my opinion) messy relationship and derives some binary conclusions about the preponderance of hard-bedded rheology. At 6 km data points, then binned to derive the traction-velocity relationship, the impact of the fastest flowing regions is muted.

**Response:** We really appreciate the core of this criticism. The challenge of working with big data sets is distilling the data into something meaningful while still properly acknowledging the variability and limitations which is what we are striving to achieve in the manuscript. However, we strongly feel conclusions can be drawn from the data and analysis, and further, that those presented in the manuscript are well supported. The current state of knowledge in Greenland is that there is no relationship between velocity and traction for the fast-flowing marine terminating glaciers as found by Stearns at al. 2018, and it is unclear whether those findings apply further inland and in regions without marine-terminating glaciers. Our analysis is designed to fill this knowledge gap, and further, does it in a novel way that minimizes the biases associated with unconstrained model parameters to generate velocity-traction relationships that can be confidently interpreted in terms of bed physics.

The reviewer correctly points out, there is scatter in the relationships even after averaging the traction and velocity over large grid cells. This is expected given the variety of different substrate, bed properties, and hydrologic conditions that are likely to exist under Greenland. Yet, even in the raw data, we can simply look at distribution of grid cells in catchments 3-4 and 6-8 and visually see that tractions increase with velocity. At high velocities, tractions are also high, and there are very few grid cells that could be characterized otherwise in the either the log scale or linear scale plots (Figure R3) presented below. Thus, to a first-order, and just from the raw data, we can already determine that there is a basic rate-strengthening relationship which is consistent with theoretical expectations for ice flow over hard beds and contrasts with the work of Stearns et al. 2018 which focused on marine terminating glaciers and found no relationship between the variables.

Binning the data through the velocity range by calculating the median IQR, and 90% of the detail allows us to take the analysis further by statistically determining the finer details of the traction relationship through the velocity range. Binning is a commonly used statistical approach in our field (Armstrong et al., 2017; Dehecq et al., 2019; Stevens et al., 2018) and for large data sets in general. We find that the catchments where rate strengthening was identifiable in the raw data again clearly show rate strengthening in the binned data. The small spread of the IQR indicates the bins well represent a large portion of the data giving us confidence that the binned relationships are representative of physical conditions across a large fraction of most catchments. Similarly, the binned statistics also show where there is more complexity. For instance, while the binned data show good adherence to rate strengthening in catchment 6, the IQR and 90$^{th}$ percentile indicate Mohr-coulomb-behavior. Using these statistical attributes and the high quantity of weak-bed grid cells allows us to make the more nuanced interpretation that grid cells in this catchment can reflect either rate strengthening and Mohr-Coulomb-like behavior and thus the bed rheology is best classified as mixed.

The data binning also allows us to identify important structure in the predominantly Mohr-Coulomb-like catchments (1 and 2). For example and as the reviewer points out, the raw data in catchment 2 has a large degree of scatter with no obviously discernable structure. However, once binned, we can see that there is rate strengthening with a sharp break in slope moving to higher velocities, which is consistent with Mohr-Coulomb like behavior. Thus, we can infer that the increased scatter likely reflects the velocity independence of Mohr-Coulomb physics, allowing us to make a more precise interpretation beyond that there isn't a relationship between the variables from looking at the raw data.

We do not feel that our interpretations and conclusions are binary. The interpretations in section 2.4 acknowledge the data variability and are robust in the sense that they rely on several different aspects of the data; model fits, characteristics of the binned relationship, distribution of grid cells, data scatter, differences in the inversions, and catchment-specific preexisting knowledge. They also identify uncertainty using the spread in the SIA, SSA, and FS, as well as the binned statistics. Further, the paper is filled with qualifiers (*section headings for 2.4.1 - 2.4.4*) and nuance (*Section 4.3*) which were purposefully put in to indicate we are not drawing sharp binary conclusions from the dataset. However, we feel our results and analysis well support our mainly conclusion which is summarized in the manuscript title "Hard-bed physics mainly dictates basal traction over grounded regions of Greenland". We are not claiming hard-bed physics applies everywhere, but except for the NEGIS and some tidewater glaciers, we mostly find behavior consistent with rate strengthening. This is an import "fuzzy constraint" which can be useful for ice sheet modelers and hopefully serve as a starting point for future investigations that look at bed properties across smaller spatial scales. However, to address this comment in general, we changed the text in several locations throughout the manuscript where our language was too definitive (highlighted in yellow, lines 17, 426, 472, 482, 512). We also added text to the conclusions to make clear that our findings reflect the relationships where the velocity and traction are averaged over large scales. This serves as an important starting point for future work to look at location specific bed properties and traction laws across finer scales (Lines 514-516).

[Figure]

**Figure R3:** Same as Figure 4 in the main text except on a linear scale. The velocity – traction relationship for the SIA, SSA, and Full Stokes inversions for all eight drainage catchments. Transparency is an indicator of data density, where opaque areas indicate four or more grid cells occupy the same marker area on the plot. Magenta markers show the median traction values for 20 logarithmically spaced velocity bins, the vertical bars show the interquartile range, and the black dots indicate the middle 90 percent of the data. Black line shows a power law model fit to the binned data. No models were fit to the stokes relationships due to the velocity range (> 100 m/yr) which is limited in our analysis.

**Comment:** - The assumption (section 2.2.1) that basal drag can be approximated by the driving stress works for the interior but really falls apart near the margins. As a result, I think the limit of this analysis precludes a lot of the faster flowing regions. Given that shortcoming, the overall conclusions are weaker.

**Response:** The strength of using three different traction inversions is it allows us to infer where the driving stress may not be a good approximation for the traction. The applicability of the driving stress depends on the complexity of the flow field and the basal topography, and thus, this approximation does not necessarily fall apart near the margins. We find close correspondence between the SIA, SSA, and FS solutions up to relatively high velocities (~ 300 m/yr) in most catchments 5-8 and at even higher velocities in catchments 3-4, suggesting the SIA is a reasonable approximation at relatively high velocities. However, the hierarchical approach does highlight velocities where there are large differences between the inversions where the driving stress might not be a good approximation for the traction. This divergence is most prominent in the higher velocity zones of catchment 7 and 8. Our conclusions and interpretations of these catchments (*section 2.4.1*) account for this uncertainty.

- While it is done frequently, it is incredibly circular to prescribe a sliding relation to estimate basal drag and then use the calculated basal drag to show a correlation with velocity (section 2.2.2 and 2.2.3). The whole justification for this is described in line 137: While a linear sliding law is used for the inversion, the traction field is shown to have little sensitivity to the choice of sliding law (Joughin et al., 2004)." This paper uses the control method exclusively for the Ross Ice Streams (no basal drag and high velocity), which is opposite to how it's being used here.

**Response** Thank you for your comment. A choice of sliding law is necessary to numerically relate the traction and velocity to solve the force balance at each location for the SSA and FS inversions. We can see how it would seem mathematically circular to do so. Below we explain why that is not the case, and further why our large-scale averaging diminishes the role of any model – specific parameters.

Using the control method, the basal shear stress is estimated to maintain the stress equilibrium at each location and converges on the same value regardless of the form of the sliding relationship applied. The only assumption is that basal drag and sliding velocity are co-linear and thus can be related by a free parameter Beta. The inversion leads to spatial fields that allow a close match with the velocity observations while maintaining the force balance. Since both velocity and traction are both solutions of the inversion the logic is not circular. (Joughin et al., 2004) demonstrated this for a synthetic case: "While we have described three parameterizations for $\tau_b$ (viscous, plastic, vector plastic), inversion with any of these parameterizations proceeds roughly in the same manner and yields the basal shear stress that provides a force balance consistent with the model given by equation (1). Force balance is achieved regardless of the assumed bed rheology. As a result, even though equation (3) applies to a viscous bed, $\beta^2$ is a free parameter, so force balance can be achieved even if the bed exhibits weak plastic rather than linear viscous behavior." This applies whether there is a Coulomb or Weertman-type glacier bed thus the glaciologic setting does not affect the derived velocity - traction relationships. The velocity – traction relationships we present in the paper demonstrate this. We find both Coulomb and Weertman-style non-linear velocity-traction relationships using a linear sliding relation to invert for the traction field.

Even though the traction solution is constrained by the momentum balance, at fine spatial resolution (approximately ice thickness and below) choice of sliding law combined with the choice of regularization (which dictates how smooth the solution should be) can produce some small-scale differences in the traction pattern (Habermann et al., 2013). However, since we are averaging over large grid cells these parameter choices have a minimal effect on the traction pattern since the basal traction is almost entirely balancing the gravitational stress.

Other comments:

**Comment:** Line 10: "three different methods" but with the same assumptions and workflow. It would be clearer to say models with three different levels of complexity. This comes up again in line 83ish. Having three models that are based on the same assumptions converge does not necessarily yield confidence in the velocity-traction relationship – just highlights where the complexity is unimportant.

**Response:** Thank you for the comment. We agree with the reviewer, a hierarchy with different complexity is an excellent way to frame the manuscript and it will be adopted in the introduction of the revised manuscript (lines 77-79). To clarify, only the SSA and FS are derived using similar methods. The SIA is calculated independently and directly from the data as per section 2.2.1. Further, with the large grid cells used in our analysis the ice the traction predominantly balances the gravitational driving stress at each grid cell. Since the gravitational stress is set by the geometry of the ice sheet there is no reason the tractions are likely to be biased where all three inversions converge. However, the assumptions on the momentum balance are quite different for the SIA, SSA, and FS and thus, the differences between the velocity – traction relationships can be assumed related to those assumptions.

**Comment:** Line 46: "bypassing" is too strong a word. We have some models that can be used to infer basal processes, but they are really far from capturing details from localized observations at the ice bed interface.

**Response:** Ok, we agree and have changed the text to. "Improvements over the last decade in satellite observations of surface velocity and geometry have made it possible to infer bed properties across ice thickness length scales based on ice dynamics (Gillet-Chaulet et al., 2016; Habermann et al., 2013; Minchew et al., 2016)." (Lines 44-45)

**Comment:** Line 76: How can rheology have impacted the findings of Stearns and van der Veen? This is a fairly bold statement; if you keep it in it needs justification. I think it's ok to say it's still under debate and leave it there.

**Response:** We agree and have removed the comment on rheology and made the change as suggested (see lines 73-75).

**Comment:** Line 254: What is meant by "visually well defined" and "show good agreement with the model fit at both high and low velocities". I thought the model was fit to these binned points.

**Response:** The reviewer is correct the model is fit to the binned data. We are simply pointing out that relationship in the binned data is clear, and that fitting a power-law to these bins provides a reasonable fit. Since the following comment address the same section, the revised text is presented together in the next response.

**Comment:** Line 255: High $R^2$ (.22 -.99)? It's more representative to specify where it's high and where it isn't - especially since it's closer to .99 in most places. Also specify what the $R^2$ value refers to (binned or raw data).

**Response:** This is a good idea and ties into our interpretations in section 4.2 (lines 258-263). We will change the description of the $R^2$ text to make it more specific as follows:

"The relationship between the binned variables (magenta dots) is visually well defined in all catchments and for each estimate of traction. In general, there is good agreement with the binned data and model fit at both high and low velocities. The $R^2$ is high in catchments 3, 4, and 6 through 8 [0.88 – 0.99] indicating

that the power-law model generally describes most of the variability of the binned relationships (Figure 4, supporting Table S1) for the SIA and SSA tractions. In the remaining catchments (1, 2, and 5) the R2 is lower [0.22 – 0.69], which results from the increasing curvature in the binned data at higher velocities which is not captured with the power law..”

Figure 5: Add labels of catchment to each subplot for color-blind folks. Is the grey line the cdf of the binned velocity data or the input data? I'm guessing the latter, which is misleading because the velocity-traction curves are based on the binned data. Don't these graphs also show that the models diverge at fast flow (I didn't like framing the models this way at the start, but since you did it tshould be revisited).

**Response:** The reviewer is correct that the CDF is for the actual data distribution and not the binned data. This was a mistake on our part which we correct on the updated figure. We also add PDFs to better illustrate the data distribution and indicate that the data is sparse at high velocities, and additionally add figure labels to each plot as suggested.

The reviewer is also correct that the models do diverge in some catchments, with the most prominent divergence found in catchment 7 and 8. This divergence suggests the traction relationship could be either rates strengthening or Mohr-Coulomb-like at high velocities. This uncertainty is addressed and acknowledged in section 4.2.1.

**Comment:** Figure 8: Doesn't the fact that strong and weak beds exist roughly equally in slow and fast flow regions show that bed strength is not related to velocity? If so, then a sliding relation that equates the two falls apart.

**Response:** Since weak and strong beds are defined relative to the ice-sheet wide relationship through the entire velocity range, by this definition, we should expect a similar proportion of weak and strong beds in slow- and fast-moving regions. As such, we agree with the reviewer that it was confusing to put the distribution of weak and strong beds from fast- and slow-moving regions on a prominent spot in the figure. We removed this from Figure 8 and took out the associated text. However, on a physical level, weak-bedded slow-flowing regions can occur in cavitation range of hard-bed sliding (Gagliardini et al., 2007). This is the region of increased curvature before reaching $CN$ on the black curve of Figure 3. Here, you can have lower traction for a given sliding speed compared to a Weertman relationship while still locally balancing the gravitational stress.

**Additional Comments from the Editor**

**Response:** McCormack et al. 2020 presented a method to optimize the driving stress length scale by comparing the direction of the velocity and driving stress vectors. When the direction difference is minimized, it can be assumed that the length scale is sufficient where effect of stress gradients on the flow field are minimized. This is a different, but highly analogous approach to our optimization method which compared the difference SIA and SSA presented in section 2.3. Even though they are similar, our method has a distinct advantage in situations where longitudinal stress gradient coupling dominates the higher-order stresses. Here, the driving stress and flow vectors could still be aligned with significant longitudinal stress gradients, and thus incorrectly suggest the influence of stress gradients is minimized at a given length scale.

Following their study, we also compared the direction of the driving stress vector to the velocity for different filter sizes (Figure R4). We find the RMSE difference in the driving stress and velocity vectors is probably best minimized at 9 km over the entire study area.  However, the optimal length scale for our analysis also took into consideration the spatial resolution near the margin and total velocity range over which the velocity and traction could be compared. This is important since many of the defining flow characteristics of the traction laws presented in Figure 3 are expected to occur at high velocities which are found near the margins. Thus, we chose 6 km for analysis, but also included our analysis done with 9 km in the supplemental (Figure S3) which would be interpreted almost identically in terms of predominate bed physics in each catchment.

We note that our surface elevations were smoothed using a Gaussian filter which was found to generally outperform the triangular and bilinear filters tested in McCormack et al. 2020. While we use a spatially

[Figure]

invariant filter in our analysis, McCormack et al 2020, found that beyond $4H$, they performed similarly to filters that scaled with ice thickness.

**Figure R4** – Comparison of grid scale to the RMS error of the driving stress vector direction compared to the velocity vector direction. Dashed lines show same analysis for different subsets of the data (margin regions < 1200 m.a.s.l. – yellow line, interior regions > 1200 m.a.s.l. – grey line, thickness > 1500 m – magenta line, thickness < 1500 m – cyan line).

**Comment:** 2. In regard to the *likely* basal thermal state, given the rigor with which the authors consider the uncertainties in other elements, I was surprised that the authors did not consider two alternative scenarios: A. The uncertain region is frozen. B. The uncertain region is thawed. Perhaps only the latter scenario is relevant to the methods employed by this study, but as the lead author of the basal thermal state investigation, I caution against prescribing too much confidence in our knowledge of that boundary condition yet.

**Response:** This is a good suggestion and is something we overlooked. We re-ran the analysis to include the uncertain regions of the likely basal thermal state map (Figure R5). By comparing the figure below to Figure 3 in the main text, we can see the uncertain regions are almost universally slow flowing and thus inclusions of the uncertain regions have little impact on our analysis. The exception to this is catchment 3, where the uncertain regions occupy can occupy relatively fast-flowing regions (up to ~ 80 m/a). Here, the uncertain regions have exceptionally high tractions for their velocity. This would be consistent with the uncertain regions being frozen.

[Figure]

**Figure R5 –** The velocity – traction relationship for the SIA , SSA, and Full Stokes inversions (6 km) including the uncertain regions of the basal thermal state mask. Transparency is an indicator of data density, where opaque areas indicate four or more grid cells occupy the same marker area on the plot. Magenta markers show the median traction values for 20 logarithmically spaced velocity bins, the vertical bars show the interquartile range, and the black dots indicate the middle 90 percentile of the data. Black line shows a power law model fit to the binned data. No models were fit to the stokes relationships due to the velocity range (> 100 m/yr) which is limited in our analysis.

**Comment:** 3. Aschwanden et al. (2016) used a bed rheology that depends partly on bed elevation and found that it worked well for their PISM simulations. Please consider the relevance of that study to your

investigation, and perhaps even evaluate even if there is a relation between inferred strong/weak beds and bed elevation.

**Response:** Aschwanden et al. (2016) found using bed elevation as an ad hoc way to tune the sliding relationship helped better capture ice flow and ice speeds within marine-terminating regions of the ice sheet. While not explicitly tied to physics, the justification is that in regions where the bed elevation is below sea-level, sea-level sets the basal pressures and the pressure dependency of the sliding relationship.

The distribution of weak beds and their relationship to hydrology and elevation is examined in detail in a follow up manuscript that is currently submitted (authored by Maier and Gimbert). However, since our relationships are mostly for grounded regions we wouldn't expect a strong dependence on bed elevation on bed strength over much of our study regions given the physical justification for the parameterization. There are some exceptions, the most obvious being the Humboldt and Petermann Glaciers, which have partially floating ice tongues and the Northeast Greenland Ice Stream where the area approximately enveloped by the 200 m/yr contour is below sea level. These regions are classified as weak bedded and would be consistent with the parametrization of hydrology used in the PISM experiment. The same interpretation that inspired the ad-hoc elevational relationship is suggested in our manuscript in lines 409-413:

However, for the low-velocity weak-bedded grid cells, quick comparison to the bed elevation map (Morlighem et al., 2017) indicates they are likely not related to elevation, and instead, seem to be more closely related to the onset of surface melting near the snowline in catchment 6. This is detailed in lines 431 to 438.

[revised manuscript text omitted]

---

## Author Response (AR2)

**Dear Editor,**

We have updated the manuscript to address the remaining reviewer comments (below).

Almost all of the reviewer comments are directed at the strength of our conclusions and how its conveyed in the text. We agree that the abstract and conclusions came off as too commanding in the previous version. To address this, we have rewritten the abstract, conclusions, as well as the first paragraph of section 4.2 to better convey the certainty and degree to which the basal traction is mainly dictated by hard bed physics. These sections are highlighted in yellow in the revised manuscript. In the remaining sections of the manuscript, we feel the text is sufficiently nuanced as is.

In terms of the confusion regarding fast and slow flow: To first clarify, the 100 m/yr used for inversion purposes was to delineate where the FS resolution is coarser than our 6 km grid resolution, and is thus is not physically defined boundary. For the remainder of the manuscript, we used fast and slow as relative terms in the manuscript to aid in our description of the traction relationships within each catchment (thus fast and slow are relative), and in certain places in the discussion. To address the confusion about fast-and slow-moving ice we have decided the best thing to do is to limit the use of the terms in favor of more precise quantitative or qualitative descriptions, and where they do exist, they are accompanied by quantitative description indicating the velocity where appropriate. One specific source of confusion came from Figure 7 with the weak and strong beds delineation. The previous version of the manuscript had fast-and slow- regions separated by a 200 m/yr contour. However, the purpose of the contour was mainly to visually identify concentrated areas of faster flow around the major outlet glaciers, and in the revised manuscript we identify the contour as such.

We also updated the text to redefine partially floating ice tongues as those where the flotation fraction is high, and fully grounded regions as those where sea-level doesn't influence base pressure.

Text modifications are highlighted in yellow, which are included in the end of this document. We note we also edited some minor typos/unclear phrasing, as well as labels in Figure 7 in which text was overlapping.

Please let us know if you have any further concerns,

Thank you,

Nathan Maier, Florent Gimbert, Fabien Gillet-Chaullet, and Adrien Gilbert

**Reviewer Comments:**

The authors give very thorough and thoughtful replies to our comments, which I greatly appreciate. I was a bit struck that the replies were so thorough, yet when I compare the manuscripts, there aren't nearly as many edits as I expected. I was definitely appeased by the thorough replies, but have some remaining concerns with how these comments were translated into the text. (For example, the reply to my comment about basal drag and driving stress was very thorough and highlighted the similarity between models up to "high velocities" of 300 m/yr. No edits to the text reflect this.)

I would still like to see more de-emphasis of the conclusion that this model works everywhere, even in fast flowing locations. This is true for the text as a whole, but particularly in the abstract and conclusion (which is what most people will read). Some care is taken in the text to emphasize the complexity in fast flow given that only 5% of the data cover velocities >450 m/yr, and the models diverge there. In the conclusion you state that the models work well over "grounded regions" – which is unclear to me. Aside from NEGIS and the floating ice tongues, the whole ice sheet is grounded, even the fast flowing (>1000 m/yr) regions. And, in the abstract, the issue of model performance at high velocities is simply de-emphasized with a brush-off generalization ("relationships are captured reasonably well by simple traction laws over the entire velocity range, including regions with velocities over 100 m/yr"). The last part of this sentence is really not supported by your results, which show complexity and are also data poor.

Perhaps my definition of fast flowing is just different from yours. In places (e.g. line 331) where you specify fast and slow regions, please add detail about what that means (and not just the max velocity, but the mean velocity in that region). I have a feeling in the manuscript, you're considering ~400 m/yr as fast flowing, whereas several readers will have different perspectives.

Line 510: what do you mean by "partially floating regions of the ice sheet"? Where are these regions? I suspect several of them are the high velocity (>1000 m/yr) regions that are giving me such pause.

**Editor Comments:**

I've now received two follow-up reviews from the same referees that originally reviewed your MS. Only the second anonymous referee has a few comments that I ask that you respond to prior to a final decision on this MS. They are mostly centered around ensuring that the models' nuances are suitably captured in the abstract and conclusions. The concern regarding "fast-flowing" is also compelling to me. It would perhaps help to define your boundary between slow and fast ice flow earlier in the MS and have a single definition thereof (it appears to be 200 m/yr? in other places it seems like 100 m/yr is used for inversion purposes?). For your reference, my interior-minded perspective favors 100 m/yr as the onset of fast flow, but the fact that I couldn't confidently determine what threshold you were consistently using by searching for various terms within the MS suggests that some effort toward additional coherence is needed.

[revised manuscript text omitted]